# Solubility and Valence Variation of Ce in Low-Alkali Borosilicate Glass and Glass Network Structure Analysis

**DOI:** 10.3390/ma16145063

**Published:** 2023-07-18

**Authors:** Liu Yang, Yongchang Zhu, Jichuan Huo, Zhu Cui, Xingquan Zhang, Xuanjiang Dong, Jie Feng

**Affiliations:** 1State Key Laboratory of Environment-Friendly Energy Materials, School of Materials and Chemistry, Southwest University of Science and Technology, Mianyang 621010, China; yangliu19990904@163.com (L.Y.); zhangxingquan@swust.edu.cn (X.Z.); 15700304311@163.com (J.F.); 2China Building Materials Academy, Beijing 100024, China; 13520204133@139.com (Z.C.); 18734891202@163.com (X.D.); 3Fundamental Science on Nuclear Wastes and Environmental Safety Laboratory, Southwest University of Science and Technology, Mianyang 621010, China

**Keywords:** low-alkali borosilicate, cerium, Raman, XPS

## Abstract

Low-alkali borosilicate glass was used as the immobilization substrate, and Ce was used to replicate the trivalent and tetravalent actinides, in order to create simulated waste glass through melt heat treatment. The valence of Ce and solubility of CeO_2_ in waste glass were studied as well as its network structure and thermal and chemical stability. The solubility of Ce in waste glass was examined by XRD and SEM. The network structure was examined by Raman spectroscopy. The valence of Ce was determined by X-ray photoelectron spectroscopy. Thermal analysis and product consistency (PCT) were employed to determine the thermal and chemical stability of waste glasses. The results show that the solubility of cerium in low-alkali borosilicate glasses is at least 25.wt.% and precipitates a spherical CeO_2_ crystalline phase when it exceeds the solid solution limit; Ce is immobilized in the glass by entering the interstices of the glass network. Depolymerization and the transition from [BO_3_] to [BO_4_] occurs when CeO_2_ doping levels rise. About 60 percent of Ce^4+^ is converted to Ce^3+^, and the thermal stability of glass rises then falls with the increase of CeO_2_. All samples exhibit strong leaching resistance, with the average mass loss of Ce at 28 days being less than 10^−4^ gm^−2^d^−1^.

## 1. Introduction

As a highly efficient and clean energy source, nuclear power has been vigorously developed in China. By 2020, nuclear power will already account for 4.8% of China’s total electricity generation [1]. The nuclear power industry in China uses a closed nuclear fuel cycle technology path, in which uranium and plutonium are converted into mixed oxide material during spent fuel reprocessing and then recycled as nuclear fuel [2,3,4]. The high-level radioactive waste (HLW) generated by this process contains 95% uranium, 1% plutonium, and 4% other actinides and various fission products, all of which are highly radioactive, toxic, and have long half-lives [5,6,7]. It is essential to properly dispose of HLW since it is of major concern if it penetrates the ecosystem [8]. The disposal of high-level radioactive waste in China comprises immobilization in an inert substrate and subsequent deep geological disposal in a geological repository [8,9,10]. Glass solidification is currently the only industrially available technology for treating HLW, and borosilicate glass is considered to be a good glass solidification substrate due to its good chemical durability and irradiation resistance [7,11,12]. All elements in HLW can be accommodated in borosilicate, but the low solubility of some long-lived radioactive actinides, such as Np, Pu, Cm, and Am, in borosilicate glass can limit the development of glass curing technology. Thus, it is crucial to increase the solubility of these long-lived radioactive elements in glass curing substrates [13,14,15,16].

As the energy released during the decay of radionuclides can cause irreparable damage to humans, it cannot be used directly in experimental research. Cerium was chosen to simulate the actinides for experimental studies because of the similar chemical properties of cerium and actinide. The ionic radii of Ce^3+/4+^, Pu^3+/4+^, and Np^3+/4+^ are similar (8 coordination: Pu^4+^ = 0.096 nm, Np^4+^ = 0.098 nm, Ce^4+^ = 0.097 nm; 6 coordination: Pu^4+^ = 0.086 nm, Np^4+^ = 0.087 nm, Ce^4+^ = 0.087 nm, Pu^3+^ = 0.100 nm, Np^3+^ = 0.110 nm, Ce^3+^ = 0.101) [17]. The results of Wang et al. showed that 10% of Ce could be included in iron phosphate glasses, and when doping was above the solid solution limit, monazite crystals (CePO_4_) precipitated [18]. Yu et al. showed that the solid solution limit of Ce in multiple alkaline-earth borosilicate glasses and glass-ceramic-hardened actinides did not exceed 13% [19]. In the research of Lopez et al., the solubility of Ce^3+^ in glass is higher than that of Ce^4+^; alkalinity affects the redox equilibrium of the glass melting process, and increasing the alkali oxide in the glass shifts the redox equilibrium towards a higher oxidation state, thus affecting the reduction of Ce^4+^ to Ce^3+^ [20].

In this experiment, a commercially prepared low-alkali borosilicate glass was chosen as the substrate, and Ce was used to simulate the tetravalent and trivalent actinides to investigate the structural properties of solubility and valence distribution of Ce in low-alkali borosilicate glass to provide a new glass substrate option for treating HLW or give new ideas for the development of formulations for waste immobilization.

## 2. Experiment

### 2.1. Simulated Waste Glass Preparation

A commercially prepared low-alkali borosilicate glass was used as the hardening substrate (see Table 1), and a homogeneous mixture was prepared by adding CeO_2_ (aladdin, 99.99%) at different mass fractions (10%, 20%, 25%, 30%) and grinding in an agate mortar for 30 min. A total of 25 g of the mixture was then put in a ceramic crucible and subjected to melt heat treatment in a muffle furnace (from room temperature to 800 °C for 2 h at a rate of 10 °C·min^−1^ and then to 1400 °C for 3 h at the same rate). Glass melt was poured onto a hot graphite plate for air quenching to obtain simulated waste glass after heat treating. Samples for thermal analysis were subjected to water quenching.

### 2.2. Characterization

Glass powders were examined for physical phase analysis using an X’ Pert PRO diffractometer (Panalytical, Malvern, UK) (Cu Kα (λ = 1.54056 Å) radiation at room temperature (2θ = 10–80°, 40 kV, 40 mA)). The sample morphology and the composition of the precipitated crystalline phases were analyzed by a Carl Zeiss Ultra 55 field emission scanning electron microscope (FE-SEM, Carl Zeiss AG, Jena, Germany) with energy dispersive X-ray spectroscopy in the accelerating voltage of 15 kV and a working distance of approximately 7.2 mm. Raman spectra in the 200–2000 cm^−1^ range were recorded using a CCD detector (In via, Renishaw, UK). An ESCALAB 250 Xi (1486.6 ev Al-ka radiation, 12 kv, 120 w, Thermo Fisher Scientific, Waltham, MA, USA) was used to acquire X-ray photoelectron spectra (XPS) of powder samples at room temperature to determine the content of different valence states of elemental Ce in the glass. Differential scanning calorimetry (DSC SDT Q600, TA Instruments Inc, New Castle, DE, USA) for glass powders with particle sizes less than 200 mesh was used to measure the typical temperature of glass from room temperature to 1300 °C at a rate of 10 °C/min in an air environment.

### 2.3. Anti-Leaching Experiment

A static leaching test was carried out to assess the resistance of the samples to leaching using the ASTM Product Conformance Test (PCT) method [21]. One gram of cleaned and dried glass powder, sieved between 100 and 200 mesh, and 10 mL of deionized water were placed in a Teflon vessel in a hydrothermal reactor and placed in a blast furnace set at 90 ± 2 °C for 28 d to simulate a geological reservoir environment [22]. After 28 d, the leachate was removed using a syringe with a filter device and the leachate was analyzed by inductively coupled plasma mass spectrometry (ICP-MS, 7700X, Agilent Technologies, Santa Clara, CA, USA) to determine the concentration of each element (Si, B, Ca) in the leachate, and the normalized mass loss NL_i_ (g·m^−2^) was calculated using Equation (1) [23]:(1)$${NL}_{i}=\frac{C_{i}\cdot V}{f_{i}\cdot S\cdot ∆t}$$
where V is the volume of the leachate (m^3^), S is the exterior area of the sample powder (m^2^), and C_i_ is the percentage of the element i in the leachate. F_i_ is the mass fraction of the element i in the sample. S/V is the surface-area-to-volume ratio of glass powders, which was chosen to be 2000 m^−1^ based on the standard. ∆t represents the test duration.

## 3. Results and Discussion

### 3.1. Physical Phase and Microstructural Analysis

Figure 1 shows the XRD diffraction patterns of Cx (x is the wt.% of CeO_2_ in simulated waste glass) series samples, and it can be seen that all samples have a wide hump in the diffraction angle θ = 20~34° range, which is due to the process-ordered structural characteristics of the amorphous glass. When x ≤ 25, the hump indicates that the entire sample is in a homogeneous glassy state, and Ce is well contained in the glass network. Sharp diffraction peaks begin to develop at x = 30, and as the value of x grows, the intensity of the peaks increases. This indicates that when the mass percent of CeO_2_ is doped to 30%, the crystalline phase begins to precipitate in the waste glass and the amount of precipitated crystalline phase grows as the extent of doping increases. By comparing with the standard PDF card data from the International Centre for Diffraction Data (ICDD), the diffraction peaks of the precipitated crystalline phase were in agreement with the characteristic diffraction peaks of cerium squared (CeO_2_; PDF#78-0694) [19]. It is shown that the loading of CeO_2_ in low-alkali borosilicate glasses is at least 25 wt.%, and the precipitated crystalline phase of cerium squared is formed when the solid solution limit is exceeded.

### 3.2. SEM and EDX Analysis

The backscattered scanning electron microscope (SEM) plots and the energy dispersive X-ray spectrometer (EDX) spectra of the cross sections of the simulated waste glass having different levels of CeO_2_ doping are displayed in Figure 2a–d correspond to the simulated waste glasses with CeO_2_ doping of 10%, 20%, 25%, and 30%, respectively. Significantly, homogeneous glassy phases with no second phase are present in Figure 2a–c. In Figure 2d, in addition to the glassy phase, a new spherical second phase appears, and there are depressions in the interval between the two phases, demonstrating that liquid–liquid phase separation occurred during the glass melting process. Figure 2e illustrates the results of scanning the EDX spectrum at position f in Figure 2d to identify the precipitated second phase. The second phase is primarily composed of Ce and O, with a minor amount of Si (likely a small bit of glass phase superposed on the top), so the precipitated second phase is judged to be CeO_2_. The results show that when the doping amount of CeO_2_ is less than or equal to 25%, the solidified body is a homogeneous glass phase. When the doping amount is 30%, the CeO_2_ crystalline phase is precipitated. This is in agreement with the X-ray diffraction results.

### 3.3. Raman Analysis

Raman spectra are scattering spectra that are created when molecules or atomic groups vibrate or rotate under the influence of potent monochromatic light, causing an energy level jump. Through Raman spectroscopy, information can be obtained regarding the unique spectra of each molecule or atomic group in the sample, as well as changes in the short-range ordered structure of the sample [24]. Therefore, the effect of CeO_2_ on the change of network structure of low-alkali borosilicate glass can be analyzed by Raman spectroscopy. Figure 3 shows the Raman spectra of C_0_, C_10_, C_20_, C_25_, and C_30_. As can be seen in the figure, the C_30_ Raman spectrum exhibits a characteristic scattering peak at 467 cm^−1^, which corresponds to CeO_2_, according to the literature [18]. This agrees with the findings of the XRD. The scattering peaks at 400–650 cm^−1^ are related to the bending vibrations of the Si-O-Si bonds in [SiO_4_] [25,26]. The vibrations of the Si-O-Si bonds of polyadic (>5), tetragonal, and ternary rings are 442 cm^−1^,492 cm^−1^ and 604 cm^−1^, respectively [27]. The inset (a) in Figure 3 illustrates that C_0_ scattering is predominantly located at 450 cm^−1^, indicating that most of the bending vibrations of Si-O-Si originate from the polyadic ring. The peak shifts towards higher wave numbers with the addition of CeO_2_, implying a transition from polyadic rings to tetragonal and ternary rings [28]. Because of CeO_2_, free oxygen has been introduced into the system. In the presence of an increasing amount of free oxygen, the O/Si ratio increases, and the Si-O-Si link is broken, generating non-bridging oxygen, causing the depolymerization of the glass network. The B-O-B stretching vibration in the [BO_3_] units is responsible for the band between 1300 and 1500 cm^−1^ in inset (b) and the scattering peak at 640 cm^−1^ in inset (a) [29]. The peaks associated with [BO_4_] units exhibit a significant weakening followed by a slight enhancement from 0 to 25, while those associated with [BO_3_] units exhibit an enhancement followed by a weakening, demonstrating that Ce causes a shift in [BO_3_] and [BO_4_]. The increase in [BO_3_] occurs at the expense of [BO_4_] units, when CeO_2_ doping is minimal, and the high field strength Ce^3+/4+^ breaks the B-O bond in [BO_4_] units to convert [BO_4_] to [BO_3_]. With the increase of CeO_2_, a significant amount of free oxygen is introduced from CeO_2_, and [BO_3_] combines with the free oxygen to change into [BO_4_] [3,30]:(2)$$O^{2-}+2BO_{3/2}\to 2BO_{4/2}^{-}$$

The stretching vibration of Si-O-Si in the Q^n^ units (*n* = 0, 1, 2, 3, 4, representing Si in the [SiO_4_] coupled to 4, 3, 2, 1 non-bridging oxygen, respectively) is what causes the broadband in the region of 800~1200 cm^–1^ [30,31]. To analyze the variation of various Q^n^ units more intuitively, Gaussian fit was performed for the broadband in the range of 800~1200 cm^−1^, and the fitting results were obtained as shown in Figure 4. The units of Q^0^, Q^1^, Q^2^, Q^3^, and Q^4^ are, respectively, 850–870 cm^−1^, 900–920 cm^−1^, 950–970 cm^−1^, 1050–1100 cm^−1^, and 1120–1190 cm^−1^ [29,32]. Figure 4f shows the change of Q^n^ from 0% to 25% of CeO_2_ doping. As can be observed, Q^n^ in C0 are Q^2^, Q^3^, and Q^4^, with Q^3^ making up the biggest percentage, showing that the low-alkali borosilicate glass network is mostly made up of several interconnected silica-oxygen tetrahedral groups (ternary, quaternary, and polycyclic rings). The tetrahedral group is dominated by polycycles because the Raman peaks for C0 in the 400~650 cm^−1^ range are primarily centered at 450 cm^−1^. The presence of Q^0^ and Q^1^ units as well as a considerable decrease in Q^3^ units after the addition of CeO_2_ demonstrate the disruption of the glass network structure and depolymerization. Q^0^, Q^1^, and Q^2^ increase smoothly from C_10_ to C_30_, but Q^3^ exhibits a slow-to-fast drop. When the doping of CeO_2_ increases from 10% to 20%, there is barely any difference in Q^3^, but when it increases from 10% to 20%, Q^3^ dramatically decreases, indicating that silicate network structure is stable when CeO_2_ doping is ≤20% and Ce is immobilized in the glass matrix by entering the silicate network gap. As the degree of polymerization grows from Q^0^ to Q^4^ and the resistance to Ce ions leaving the network structure increases, Ce can be effectively immobilized in the glass with more Q^n^ (*n* = 3,4) [33,34]. The network gap accommodates an increasing number of Ce ions as the Ce doping level rises, and the Si-O link is broken by the strength of the ionic field, which significantly reduces the degree of polymerization. When the doping level hits 30%, Ce separates from the network structure to form a CeO_2_ crystal phase. In conclusion, Ce is effectively immobilized in low-alkali borosilicate glasses because the [SiO_4_] units in the network are dominated by Q^3^.

### 3.4. Valence Analysis of Ce

Ce has two chemical valence states (+3 and +4) in glass melt. The solubility of Ce in glass differs from different valence states. According to the literature, the solubility of Ce^3+^ is higher than Ce^4+^, so it is essential to analyze the valence composition of Ce in glass melt to understand the higher solubility of Ce in low-alkali borosilicate glass. During this experiment, the valence composition of Ce was analyzed by XPS. Figure 5 shows the XPS spectra of samples with different CeO_2_ doping amounts. Fitting 10 peak data of the Ce 3d XPS spectra and the ratios of Ce^3+^ and Ce^4+^ of C_10_, C_20_, C_25_, and C_30_ are shown in Table 1; +3 and +4 valence states of Ce are present in all samples, as shown in Figure 5. The fitted peaks of the Ce 3d spectra consist of six peaks belongs to Ce^4+^ (ν_0_, ν_1_, ν_2_, ν_0′_, ν_1′_, ν_2′_) and four peaks belonging to Ce^3+^ (u_1′_, u_0′_, u_1_, u_0_). Due to the coupling of the orbital and spin motions of the electrons, there is spin-orbit splitting (SOS) in the Ce 3d orbitals. Six peaks belonging to Ce^4+^ (ν_0_, ν_1_, ν_2_, ν_0′_, ν_1′_, ν_2′_) and four peaks belonging to Ce^3+^ (u_1′_, u_0′_, u_1_, u_0_) make up the fitted peaks of the Ce 3d spectra. Spin-orbit splitting (SOS) occurs in the Ce 3d orbitals as a result of the electrons’ orbital and spin movements being coupled. The top of the diagram shows five peaks that are part of the 3d_3/2_ spin-orbit splitting doublet, whereas the bottom five peaks (u_1_, u_0_, v_0_, v_1_, v_2_) are part of the 3d_5/2_ spin-orbit splitting doublet [17,35,36]. The respective peak areas of Ce^3+^ and Ce^4+^ were used to calculate the relative concentrations of the two valence states, and the findings are displayed in Table 2. The Ce^3+^/Ce^4+^ content ratio was close to 1.43 when the percentage of doped cerium was 10% or 30%, while the Ce^3+^/Ce^4+^ content ratio was slightly higher than 1.6 when the concentration of doped cerium was between 10% and 30%, which is higher than the results of Yu and Tong et al. [19,28]. Increasing the alkalinity of the melt biases the redox equilibrium towards a high oxidation state. Therefore, cerium is more soluble in low-alkali borosilicate glasses than in conventional borosilicate glasses.

### 3.5. Thermal Stabilization

The disintegration of radionuclides produces heat and raises the temperature of the curing body when waste glass is subjected to deep geological treatment. High temperature can cause crystallization of the curing body, and the volume of the curing body may be changed, which can affect the chemical durability and mechanical properties of the curing body. Therefore, in addition to a high waste-loading rate, good thermal stability is also required for the curing body. The thermal analysis of the samples with varied CeO_2_ doping levels are shown in Figure 6. It is concluded that the glass transition temperature (T_g_) of the simulated waste glass is higher than 800 °C and tends to increase and then decrease with the increase of CeO_2_.

The fact that Ce enters the glass network structure by forming Si-O-Ce bonds, which increases network connectivity, is likely what causes the increase in T_g_. Additionally, a stronger Ce ionic field can prevent other ions from moving freely, requiring more energy to complete structural rearrangement. The decrease of T_g_ may be attributed to the introduction of large amounts of free oxygen with the incorporation of excess Ce, causing the glass network to become highly depolymerized [37,38]. In general, the precipitation temperature of the glass, T_c_ ≈ T_g_ + 50 °C, and the temperature of the cured body during deep geological treatment does not usually exceed 500 °C [39]. Therefore, the simulated waste glass cured bodies made from low-alkali borosilicate have good thermal stability.

### 3.6. Anti-Leaching Features

The product consistency method (PCT) was used to assess the samples’ resistance to leaching from C_10_ to C_30_ [21]. After the 28 d test, the normalized leaching rates of Si (NL_Si_), B (NL_B_), Ca (NL_Ca_), and Ce (NL_Ce_) were determined, as shown in Figure 7. All samples were in line with the requirements of the Chinese nuclear industry standard [40]. The normalized leaching rates of Si and B were 3.37 × 10^−4^~4.65 × 10^−4^gm^−2^d^−1^ and 5.26 × 10^−4^~6.99 × 10^−4^gm^−2^d^−1^, respectively, and the normalized leaching rates of Ca were 1.05 × 10^−3^~1.12 × 10^−3^gm^−2^d^−1^ lower than those of basalt and multi-alkaline earth borosilicate glasses. Ce was not detected in C_10_, C_20_, and C_25_, and the normalized leaching rate of Ce in C_30_ was 2.3 × 10^−5^ gm^−2^d^−1^. In summary, the simulated actinide curing bodies prepared from low-alkaline borosilicate glasses have good chemical durability.

## 4. Conclusions

A commercial low-alkali borosilicate glass substrate was used in this paper for the preparation of simulated waste glass through melt-heat treatment, and XRD, SEM, Raman spectroscopy, XPS, thermal analysis, and PCT were used to analyze the structure and features of the samples, leading to the following conclusions:-The solubility of simulated trivalent actinide nucleophile Ce in low-alkali borosilicate glasses is ≥25%, and spherical CeO_2_ crystalline phases precipitate when the Ce doping exceeds the solid solution limit.-The Raman results show that the addition of Ce causes glass network depolymerization, and Ce is retained in the glass matrix of low-alkali borosilicate glasses by adsorbing into the gaps in the glass network. Furthermore, Ce causes a shift between [BO_3_] and [BO_4_].-In low-alkali borosilicate glasses, more than 50% of Ce^4+^ is converted to Ce^3+^, and the low-alkali borosilicate glass has good thermal stability and resistance to leaching for immobilizing simulated actinide Ce.

## Figures and Tables

**Figure 1 materials-16-05063-f001:**
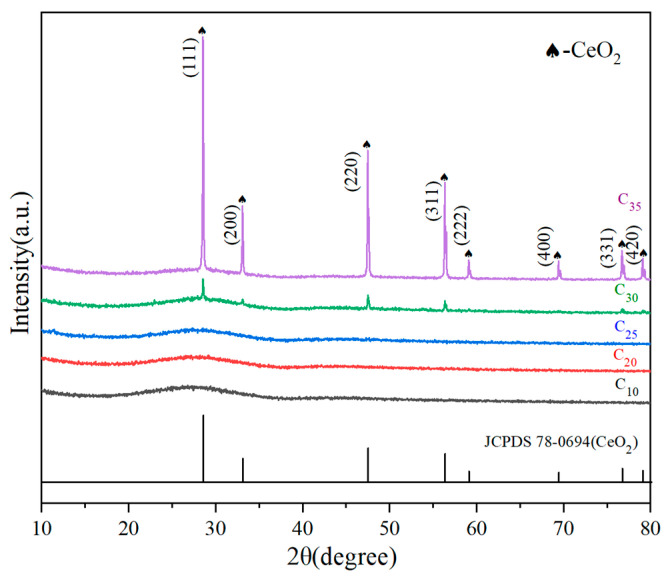
XRD diffraction patterns of C_x_.

**Figure 2 materials-16-05063-f002:**
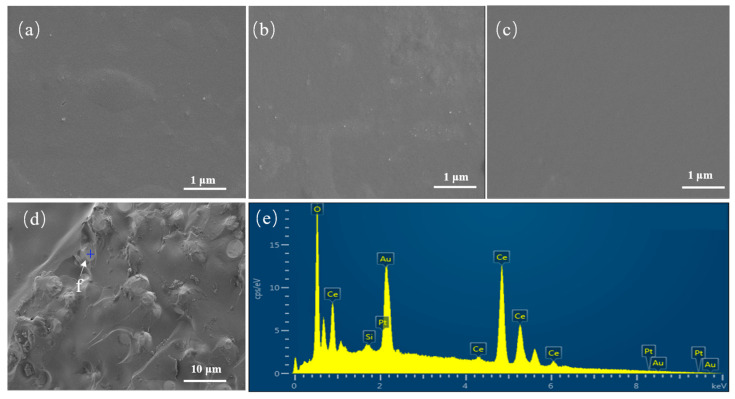
(**a**–**c**) SEM figure of C_10_, C_20_, C_25_, C_30_; (**e**) EDX spectra of point f in (**d**) image region.

**Figure 3 materials-16-05063-f003:**
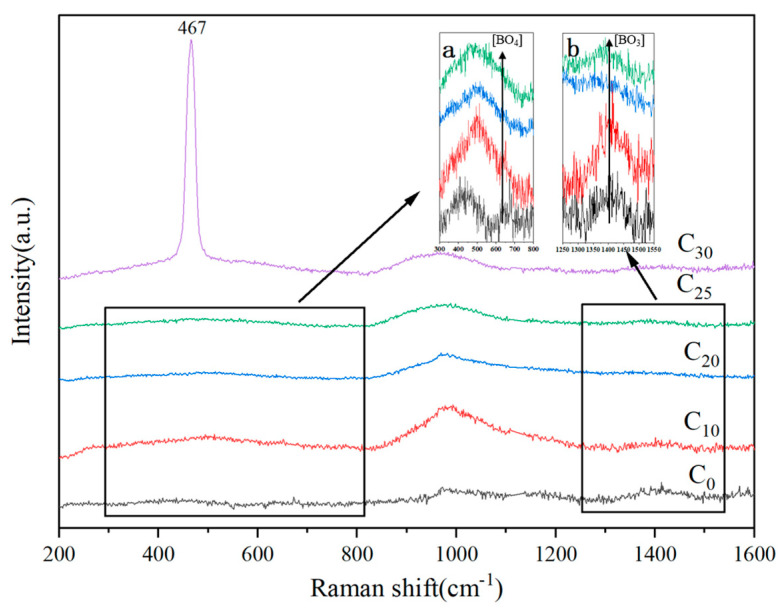
The Raman spectra of C_0_, C_10_, C_20_, C_25_, and C_30_.

**Figure 4 materials-16-05063-f004:**
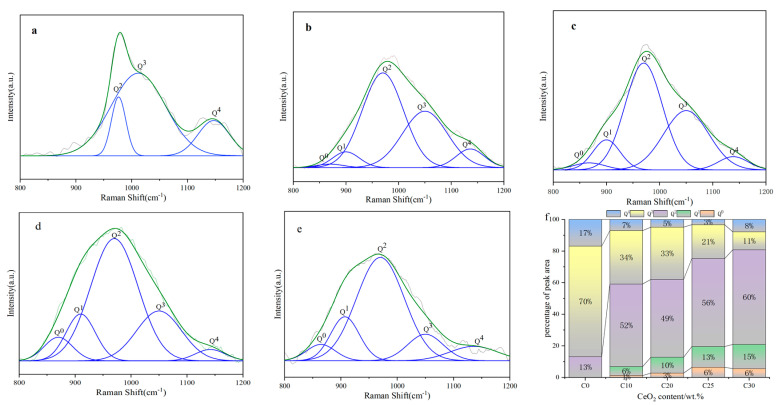
(**a**–**e**) Deconvolution map; (**f**) variation of different types of Q^n^ units from 0% to 25%.

**Figure 5 materials-16-05063-f005:**
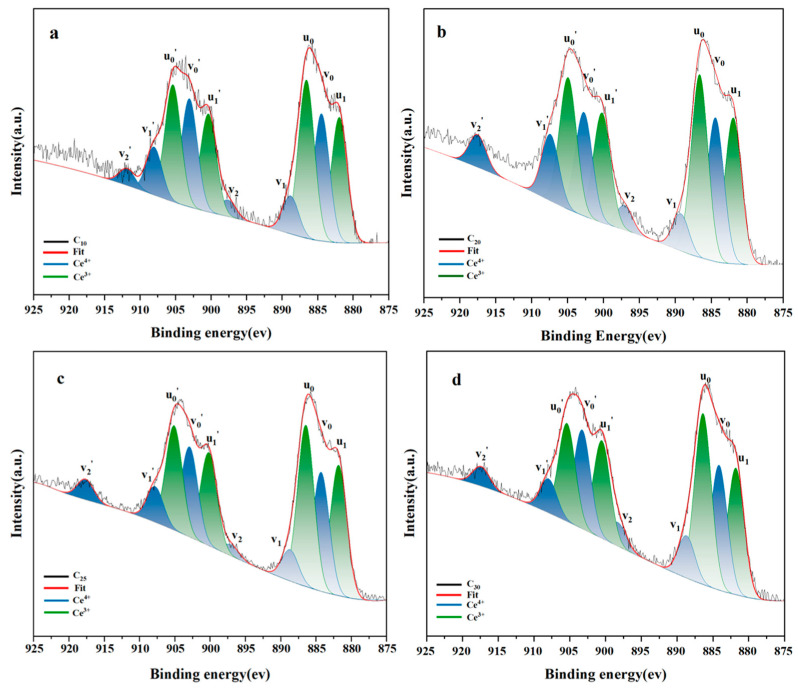
Results of XPS spectra fitting with various doped CeO_2_.

**Figure 6 materials-16-05063-f006:**
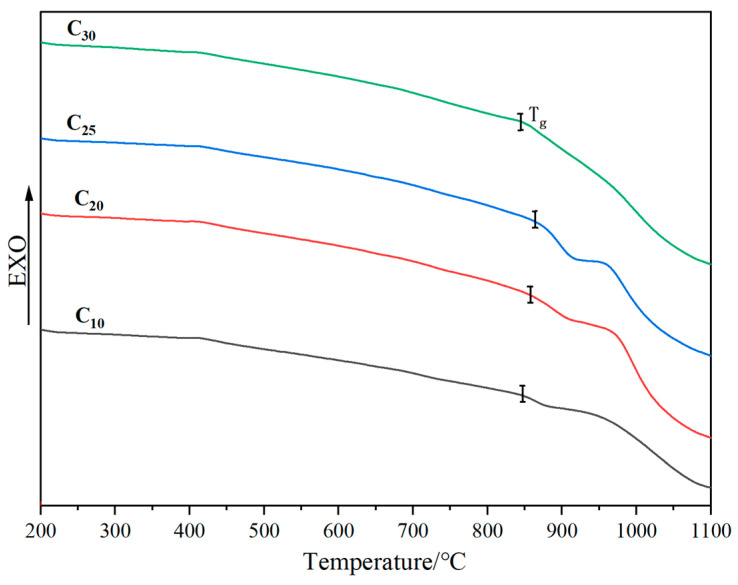
Thermal analysis of the samples with different CeO_2_ doping levels.

**Figure 7 materials-16-05063-f007:**
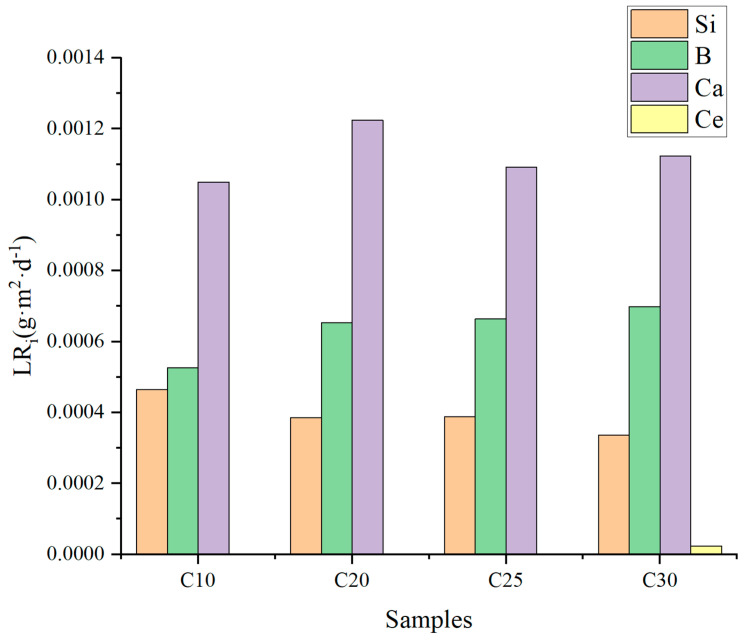
Normalized mass losses of Si, B, and Ca of C_10_~C_30_ samples.

**Table 1 materials-16-05063-t001:** Components of the low-alkali borosilicate glass.

Components	SiO_2_	Al_2_O_3_	B_2_O_3_	CaO	MgO	Fe_2_O_3_	FeO	TiO_2_	K_2_O	Na_2_O
Mass fraction(%)	54.43	14.41	6.57	22.57	0.64	0.34	0.12	0.33	0.23	0.36

**Table 2 materials-16-05063-t002:** Data of the 10 fitted peaks of the Ce3d orbital XPS spectra and the ratios of Ce^3+^ and Ce^4+^ for C10, C20, C25, and C30.

Sample	C_10_	Area	C_20_	Area	C_25_	Area	C_30_	Area
u_1′_	900.33	11,761.62	900.28	19,172.5	900.19	23,900.06	900.54	20,066.49
u_1_	881.92	15,709.68	881.86	25,657.34	881.8	33,547.52	881.83	26,054.39
u_0′_	905.41	13,709.63	904.76	24,776.7	905.11	26,859.23	905.42	20,155.53
u_0_	886.68	19,682.76	886.55	32,886.86	886.44	41,498.19	886.41	35,150.47
v_0′_	903.05	12,777.1	902.58	12,789.6	902.92	23,148.48	903.29	20,306.55
v_1′_	908.15	5146.83	907.39	10,236.81	907.78	9193.86	907.97	6999.5
v_2′_	918.72	2246.95	917.46	2967.11	917.65	5282.2	917.41	3978.63
v_0_	884.49	16,176.4	884.34	25,356.57	884.29	30,577.06	884.17	25,644.03
v_1_	889.19	4481	888.95	6740.43	917.65	5282.2	888.7	8711.89
v_2_	897.14	2162.72	896.98	2960.69	897.08	2863.49	898.31	4887.07
A		1.42		1.68		1.65		1.44
B		0.59		0.63		0.62		0.59

Note: A = Ce^3+^/Ce^4+^; B = Ce^3+^/(Ce^3+^+Ce^4+^).

## Data Availability

Not applicable.

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
