# Peer review of "Solubility and Valence Variation of Ce in Low-Alkali Borosilicate Glass and Glass Network Structure Analysis"

_materials, 2023, doi:10.3390/ma16145063_

Round 1
Reviewer 1 Report
In the article "Solubility and valence variation of Ce in low alkali borosilicate glass and glass network structure analysis " the structural and thermal properties of borosilicate glasses for different concentrations of CeO2 are presented. However, some suggestions and corrections are required for publication in Materials, which are as follows:
1 - In the Introduction, mention about the possible technological applications of alkaline borosilicate glass with the addition of CeO2 that can be used.
2 - In Figure 01, identify in the diffractogram the crystal planes referring to the diffraction peaks.
3 - In Figure 2(d), the formation of spherical particles is observed, probably CeO2 crystals, which would be in agreement with the results of X-ray diffraction. Would it be possible to determine the size of these particles as a function of CeO2 concentration?
4 - Organize table 02 to facilitate visualization.
5 - Review the text, as there are sentences that begin with a lowercase letter and the units of measurement are written incorrectly. In the text, the abbreviations EDX and EDS are shown, which would be the correct expression?
Review the text, as there are sentences that begin with a lowercase letter and the units of measurement are written incorrectly. In the text, the abbreviations EDX and EDS are shown, which would be the correct expression?
Reviewer 2 Report
The manuscript reports on the solubility of CeO2 in low alklai borosilicate glass; the results are interesting; however, the manuscript needs MAJOR corrections and modifications:
- The introduction is concise, which is a positive point. However, the aim of this study, lines 50-62, is not clear? The introduction does not present why this study is important and how it improves the understanding on the waste immobilization?
- The experimental processes are described clearly with just enough details. Probably, providing the experiments could help readers to reproduce the results, but there is no need to change this part.
Results/discussion:
1. The main issue with the obtained results is the solubility of CeO2? the residual CeO2 was due to the residual CeO2 ( not reacted during melting) or precipitated upon quenching the glass? this requires better discussion and if necessary additional experiments.
2. Figure 2: the contrast of BS images, given the difference in Z number of elements ( BS and CeO2) is different from what was expected.
3.line 130-131: how could the phase separation ( liquid separation) can be concluded from the results?
Raman and XPS results: although the results are well presented and interesting, they are not discussed in detail? the discussion is limited to fitting the bands and interpreting the Q series ...however, the origin of the changes in the Raman bands needs to be discussed. Moreover, correlations between the glass structure and crystallization as well as leaching need to be discussed... it might help the leaching behavior.
There are few mistakes ( typos and missing punctuation marks).
Round 2
Reviewer 2 Report
The authors made the necessary corrections and changes and the manuscript can be now published.